# Cactus Cladodes *Opuntia* or *Nopalea* and By-Product of Low Nutritional Value as Solutions to Forage Shortages in Semiarid Areas

**DOI:** 10.3390/ani12223182

**Published:** 2022-11-17

**Authors:** Michelle C. B. Siqueira, Juana C. C. Chagas, João Paulo I. S. Monnerat, Carolina C. F. Monteiro, Robert E. Mora-Luna, Silas B. Felix, Milena N. Rabelo, Fernando L. T. Mesquita, Juliana C. S. Ferreira, Marcelo A. Ferreira

**Affiliations:** 1Department of Animal Science, Federal Rural University of Pernambuco, Dom Manoel de Medeiros Street, Dois Irmãos, Recife 52171-900, Pernambuco, Brazil; 2Department of Agriculture Research for Northern Sweden, Swedish University of Agriculture Sciences (SLU), 901 83 SE Umeå, Sweden; 3Animal Science Department, State University of Alagoas, BR 316, km 87,5, Santana do Ipanema 57500-000, Alagoas, Brazil; 4Animal Science Department, Federal University of Norte of Tocantins, Araguaína 77804-970, Tocantins, Brazil; 5Agronomic Institute of Pernambuco, Experimental Station, Sertânia 56600-000, Pernambuco, Brazil

**Keywords:** ammonia nitrogen, *Cactaceuos*, rumen pH, semiarid, volatile fatty acid

## Abstract

**Simple Summary:**

In the different livestock production systems, forage is the main feed resource. However, the availability and quality of the forage fluctuate throughout the year due to variable environmental conditions, such as temperature, humidity, location, or lack of rainfall. In semiarid regions, this fact is even more critical. The option for forage plants adapted to the semiarid climate, such as cactus cladodes, becomes indispensable for the sustainability of the systems. Nonetheless, it is necessary to combine the cactus with high-fiber-content feeds (silage, hay, and agroindustry residues, among others) to increase fiber contents in the diet to promote ideal rumen conditions. Based on the knowledge that cactus cladodes (*Opuntia* spp. and *Nopalea* spp.) are one of the most viable crops in semiarid regions, the association with a by-product rich in NDF proves to be a more feasible alternative in terms of price and availability, with the producer making the final decision.

**Abstract:**

We aimed to evaluate the effect of the cactus cladodes *Nopalea cochenillifera* (L). Salm-Dyck. (NUB) and cactus cladodes *Opuntia stricta* (Haw.) Haw. (OUB), both combined with sugarcane bagasse (SB) plus urea, Tifton hay (TH), corn silage (CS), and sorghum silage (SS) plus urea on nutrient intake and digestibility, ruminal dynamics, and parameters. Five male sheep, fistulated in the rumen, were assigned in a 5 × 5 Latin square design. The NUB provided a higher intake of dry matter (DM) and any nutrients than SS. TH provided larger pools of DM and iNDF. The OUB and CS provided a higher DM degradation. CS provided a higher NDF degradation rate. OUB provided a lower ruminal pH. Depending on the collection time, the lowest pH value was estimated at 3.79 h after the morning feeding. There was an interaction between treatments and collection time on VFA concentrations. Due to the high degradation rate, greater energy intake, less change in rumen pH, greater volatile fatty acid production, and feasibility, we recommend using cactus associated with sugarcane bagasse plus urea in sheep diets.

## 1. Introduction

Semiarid regions are characterized by irregular rainfall distribution. The historical rainfall average in these regions ranges from 550 to 600 mm/year [1]. Additionally, the rainy season is short (3–4 months), occurring intensely and over a few days, resulting in frequent and prolonged drought periods.

Long periods of drought associated with an increasing rate of pasture degradation result in a low nutrient supply in quantity and quality to herds, which is aggravated by the continuous increase in feed prices, resulting in restrictions on productivity [2]. Thus, the adverse environment combined with economic instability intensifies the necessity to adjust and structure forage support on properties, aiming for animal production system sustainability [3,4].

The conservation of roughage, such as silage and hay, is critical for the production system due to the unfeasibility to produce forage species commonly used in ruminants’ diet. However, the productivity of these crops (corn, sorghum, and elephant grass) in semiarid regions becomes a limiting factor due to water instability and high cost, resulting in a lower forage production mass per hectare. Thus, by-products from agribusiness, such as sugarcane bagasse, can represent an important fiber source for semiarid region’s production system due to their availability and low cost.

The cactus forage has been intensively studied since 1972 by research institutes in Brazil and countries such as South Africa, Mexico, Morocco, and Tunisia [5,6,7], and today it is recognized as an energy (metabolizable energy = 2.38 Mcal/kg DM) [8] and water source for livestock in semiarid regions. Among several species studied, the most used and resistant to eventual pests in Brazilian semiarid areas are the forage cactus Miúda (NUB) (*Nopalea cochenillifera* Salm–Dyck) and Mexican Elephant Ear (OUB) (*Opuntia stricta* [*Haw*] Haw).

In a data compilation [9], it was observed that, for both genotypes, cacti have low levels of CP (46 ± 13.6 g/kg DM), NDF (286 ± 56.2 g/kg DM), and EE (11 ± 27 g/kg DM). On the other hand, cacti have considerable levels of NFC (536 ± 85.1 g/kg DM) and ash (131 ± 38.0 g/kg DM).

This study aimed to evaluate the effect of cactus forage (*Nopalea* and *Opuntia*), combined with sugarcane bagasse and urea, as an alternative to traditional conserved roughages (Tifton hay, corn, or sorghum silage), on intake, fiber dynamics, and ruminal parameters in sheep.

## 2. Materials and Methods

The study was conducted in accordance with the standards of the National Council for Control of Animal Experimentation (CONCEA) and approved by the Ethics Committee on Use of Animal for Research (CEUA; License No. 069/2016). This study was carried out in Recife, Pernambuco state, Brazil, (8°1′16.74″ S and 34°57′13.44″ W), at an altitude of 4.0 m above sea level.

### 2.1. Animals, Experimental Design, and Diets

Prior to the experiment, the animals were tagged and treated with ivermectin, vaccinated against to clostridium, and supplemented with vitamin complex (ADE). Five rumen fistulated sheep with average initial body weight (BW) of 34.0 ± 3.63 kg were housed in individual pens (0.93 × 1.54 m) fitted with feeders and waterers that were assigned in a 5 × 5 Latin square design. The trial lasted 110 days, with five consecutive 22-day periods divided into 14-day adaptation and 8-day sampling periods.

The animals were fed twice a day at 0800 and 1600 h. The experimental diets were formulated in a roughage:concentrate ratio of 69.4:30.6. The experimental diets consisted of five different roughages: cactus *Nopalea cochenillifera* (L). Salm-Dyck. cladodes (*Nopalea*) + urea/ammonium sulfate (as; 9:1) + sugarcane bagasse (NUB), cactus *Opuntia stricta* (Haw.) Haw. cladodes (*Opuntia*) + urea/ammonium sulfate + sugarcane bagasse (OUB), Tifton hay (TH), and corn silage (CS; *Zea mays* L.-Agroceres^®^ AG5055) and sorghum silage (SS; *Sorghum bicolor* L. Moench-IPA SF-15) + urea/ammonium sulfate. Both cactus genotypes are spineless. The concentrate feeds used in the diets were ground corn, soybean meal, and mineral mixture.

The cacti were harvested manually and stored in a warehouse. The cacti were processed by using a crusher MC-1001N (LABOREMUS, Paraíba-Brazil), resulting in the material being processed into a cactus soup. The “soup” was offered *in natura* to the animals, together with the other TMR ingredients. The cactus was crushed daily, prior to each feeding time. The evaluation of DM content of cactus was performed weekly to adjust the amount of feed allowed to the animals. The mixture of ingredients was performed manually in the feeders, highlighting that the spineless cactus mucilage allowed a uniform aggregation of urea. The diets were offered ad libitum as a total mixed ration (TMR), and the orts were controlled and ranged from 9 to 11% based on the total DM offered, avoiding sorting.

The chemical composition of ingredients, cactus carbohydrates composition, and ingredients proportion, and the chemical composition of the experimental diets are presented in Table 1, Table 2 and Table 3, respectively. The diets were formulated to be isonitrogenous and to attend the nutritional requirements of sheep at 25 kg BW with an average daily gain of 150 g/day [10].

The particle sizes of the sugarcane bagasse, Tifton hay, and silages were about 12.4, 12.7 and 13 mm, respectively. Then the average particle size of the preserved feed was 12 mm.

### 2.2. Data and Sample Collection

Feed and orts were weighed daily throughout the experimental period for the calculation of nutrient intake. From the 16th to 18th day of each experimental period, we performed the total feces collection to estimate the total apparent digestibility of dry matter and its constituents, using collecting bags attached to the animals’ bodies. The digestibility of DM, OM, CP, and NDF was estimated by the relationship between intake and the amount of the respective nutrients excreted in the feces, divided by the intake of the specific nutrient.

On collection days, feed, orts, feces, and the solid and liquid phases of the ruminal digest were sampled and stored in plastic bags at −20 °C. At the end of the experiment, the samples were oven-dried at 60 °C for 72 h and ground to pass through a 2 mm mesh for in situ ruminal incubation and through a 1 mm screen for further chemical analyses.

On the 19th day of each experimental period, ruminal fluid was collected before (0 h) morning meal supply and then 2, 4, and 6 h after that. The ruminal fluid pH values were measured immediately after collection, using a potentiometer (Kasavi, Model K39-0014P, Taipei City, Taiwan). The ruminal fluid was acidified with 1 mL of sulfuric acid [11], and sub-samples (40 mL) were frozen at −20 °C for later determination of the rumen ammonia nitrogen (RAN) and volatile fatty acids (VFAs) concentrations.

On the 20th day of each experimental period, four hours after the morning meal supply, we performed ruminal emptying, and on the 22nd day, this procedure was performed immediately before feeding. The emptying to determine the ruminal rates and pools was performed according to the technique described by Reference [12]. The total weight of the ruminal digesta was registered after rumen emptying, followed by filtration through the cotton fabric to separate the solid and liquid phases. A representative sample of both phases was collected and frozen (−20 °C) for further analysis of DM, neutral detergent fiber (NDF), and indigestible neutral detergent fiber (iNDF). After sampling, the phases were mixed again, and the remaining digesta were returned to the rumen.

### 2.3. Chemical Analysis

Dry matter (DM; method 934.01), ash (method 942.05), crude protein (CP; method 968.06), and ether extract (EE; method 920.39) were analyzed according to Reference [13]. Subsequently, NDF was analyzed by using heat-stable α-amylase (Termomyl^®^, 2X), as described by Reference [14]; and corrected to ash and protein, according to Reference [15], and acid detergent fiber (ADF), as described by Reference [16]. Neutral detergent insoluble nitrogen (NDIN) was analyzed by using the Kjeldahl method [15].

Non-fiber carbohydrates (NFCs) were calculated according to Reference [17], as follows:NFC = 100% − (%ash + %EE + %(a)NDF(n) + (%CP − %CPurea + %urea).(1)

The organic matter (OM) was calculated through the difference between DM and ash contents, and the digestible organic matter was calculated as follows:DOM = OM intake × OM digestibility/1000.(2)

The rumen ammonia nitrogen concentration (method INCT-CA N-007/1) and the total N content in the urine were determined by the Kjeldahl method, according to techniques standardized by the National Institute of Science and Technology in Zootechnics (INCT-CA) [18]. The analysis of VFAs was performed by using a gas chromatograph equipped with a flame ionization detector and auto-injector equipped with a GP column (30 m × 0.250 mm, 0.25 µm; Chromosorb WAW).

The rates of ingestion (Ki), passage (Kp), degradation of DM and NDF (Kd), and iNDF (Kpi) were calculated by dividing the daily intake flow by their respective rumen pools [12].

### 2.4. Statistical Analysis

The data obtained were analyzed by using the MIXED procedure of the SAS software (version 9.2), according to the following model:Y_ijk_ = m + T_i_ + a_j_ + P_k_ + e_ijk_(3)
where Y_ijk_ is the dependent variable measured in animal j, which was subjected to the i treatment in period k; µ is the general mean; T_i_ is the fixed effect of treatment i; a_j_ is the random effect of animal j; P_k_ is the random effect of period k; and ε_ijk_ is the unobserved random error, assuming normal distribution.

All means were compared by using the Tukey test, with a critical level of 5% probability being adopted for type I error. Ruminal pH, RAN, and VFAs were analyzed as the effects of repeated measures over time.

## 3. Results

### 3.1. Intake and Digestibility of Nutrients

The NUB allowed higher (*p ≤* 0.03) DM (1024 g/d), OM (904 g/d), CP (161 g/d), and DOM (670 g/d) intakes than SS and higher NFC (433 g/d) intake than TH, CS, and SS (Table 4). The animal fed with OUB showed a lower (*p <* 0.01) NDF intake (310 g/d) than those fed with TH and CS (525 and 422 g/d, respectively).

DM digestibility was similar for all roughages, but the diet based on NUB provided higher OM digestibility (741 g/d) than other roughages. The NUB and OUB recorded higher (*p ≤* 0.01) CP digestibility (831 and 806 g/kg, respectively) compared to traditional conserved roughages. On the other hand, the TH diet provided higher (*p ≤* 0.01) digestibility of NDF (645 g/kg) compared with OUB (496 g/kg).

### 3.2. Neutral Detergent Fiber and Dry Matter Ruminal Dynamic

TH provided higher pools of DM and iNDF (593 and 178 g) than OUB, CS, and SS (408 and 121; 448 and 96.4; 421 and 101 g, respectively). The NDF ruminal pool was lower (*p ≤* 0.007) for OUB and SS than TH (Table 5).

DM and NDF ingestion rates and DM, NDF, and iNDF passage rates (g/h) were not affected by roughages (Table 5). On the other hand, OUB and CS provided a higher (*p <* 0.05) DM degradation rate than TH and SS. The higher observed value for the NDF degradation rate was for CS compared to NUB.

### 3.3. Indicators of Ruminal Fermentation

There was no interaction between collection time and roughage on ruminal pH. TH provided a higher ruminal pH than NUB and OUB. The pH showed a quadratic effect over time, presenting an estimated minimum value of 6.38 at 3.79 h after feeding (Table 6).

There was an interaction between collection time and treatment on RAN concentration. OUB and NUB presented higher (*p <* 0.01) concentration values (Table 6). Maximal RUNs of 35.4, 41.8, 17.9, and 26.5 (mg/dL) were estimated for OUB, NUB, CS, and SS at 2.02, 2.97, 3.01, and 2.87 h after morning feeding. For TH, the RUN value was maintained constantly (14.7 to 13.73 mg/dL) over collection time (Figure 1).

There was an interaction between collection time and treatment (*p <* 0.01) on VFA concentration. The animals fed with NUB and OUB showed higher VFA concentrations than those fed the other roughages (*p <* 0.05; Figure 2). There was an interaction (*p <* 0.01) between treatment and collection time on VFA concentrations. Both treatments containing cactus cladodes provided a higher concentration of all VFAs and a lower acetate:propionate ratio.

## 4. Discussion

### 4.1. Intake and Digestibility of Nutrients

The higher DM intake observed for OUB than SS may be attributed to a higher NFC content in the *Nopalea* [19]. The OM, CP, and DOM intake followed the same behavior verified for the DM.

The physical and chemical properties of feeds, such as the NDF content and moisture, can limit DM intake [20]. Sorghum silage presented low levels of DM and NFC (230 and 164 g/kg on DM) and a high content of NDF (676 g/kg on DM; Table 1) compared to the values published (280; 295 and 563 g/kg, respectively) by [21]. The low DM content in the sorghum silage is probably a consequence of the cutting in the graining rubber phase and the low NFC and high NDF contents due to the variety used, which had a low proportion of grains. This variety is the most suitable for cultivation in a semiarid region.

Forage-conservation processes, such as silage making, generate changes in the forage chemical composition. According to the intensity of these changes, the nutritional value and the quality of the forage decrease [22]. The low DM content observed in this sorghum silage may have provided inadequate lactic and acetic fermentation, thus favoring the formation of butyric acid and a strong unpleasant silage odor [23,24], contributing to the lower intake.

Another important observation was the low quality of the SS, highlighting the low levels of DM and NFC (230 g/kg as-fed basis and 164 g/kg of DM) and the high levels of NDF and ADF (676 and 469 g/kg DM; Table 1). The high content of ADF may compromise the DM digestibility, resulting in rumen repletion and decreased intake [4].

The higher digestibility of OM observed for NUB may be related to the higher intake of NFC (Table 4) associated with the lower NDF content in the diet (382 g/kg DM). The animals fed with TH showed a higher NDF intake because the TH-based diet had a higher fiber content (NDF = 546 g/kg DM).

The increase in CP’s digestibility in diets containing NUB and OUB can be attributed to the greater participation of urea + ammonium sulfate (2.0% and 1.9% of the total diet) (Table 3). Urea has a fast disappearance rate due to its high rumen solubility [25]. The TH diet presented a higher NDF digestibility than OUB, which can be explained by the lower ADF/NDF ratio (60%) of TH. The ADF content is positively correlated with the non-degradable fraction of NDF. Sugarcane has a high non-digestible fiber fraction and a low digestion rate of the potentially degradable fiber [4,26,27]. Studies with a high proportion of cacti showed low digestibility of NDF [28]. Additionally, OUB contains sugarcane bagasse, which also has low NDF digestibility.

### 4.2. Neutral Detergent Fiber and Dry Matter Ruminal Dynamic

A lower rate of DM degradation was observed for TH and SS compared to NUB, OUB, and CS. The diets with the highest proportions of NFC contained cactus cladodes, and corn silage degraded quickly in the rumen, resulting in a quicker disappearance. The NFC:NDF ratio estimated was much higher for NUB, OUB, and CS (50:50, 48:52, and 45:55, respectively) compared to TH and SS (29:71 and 36:64, respectively).

The faster NDF digestion rate of CS can be explained by the lower concentration of iNDF in this forage (Table 1). The fiber content in diets with cactus cladodes was balanced by the addition of sugarcane bagasse, a cheap and highly available ingredient. It contains high levels of NDF and iNDF and, consequently, decreases fiber digestion. Additionally, sugarcane bagasse provided more than 60% of NDF content in the NUB and OUB.

### 4.3. Indicators of Ruminal Fermentation

The average ruminal pH of 6.53 (Table 6) remained within the levels recommended as ideal for cellulolytic bacteria activity and fiber degradation (6.2 to 7.0) [29]. This is justified by the presence of physically effective fiber from roughage (sugarcane bagasse, Tifton hay, and corn and sorghum silages). The mechanisms responsible for regulating ruminal pH are physiological [30], such as saliva secretion and behavioral [31]. Diets rich in roughage provide more time spent on chewing activity, stimulating salivation. This releases buffers that neutralize the acids produced by the fermentable organic matter [32], avoiding ruminal pH changes (Table 6).

The lower pH value (6.32) verified for OUB may have been influenced by the greater amount of sugar present in the *Opuntia* (Table 2). Sugars increase VFAs’ concentration due to their high fermentation rate, and, consequently, rumen pH decreases [33]. The higher pH value (6.65) verified for TH can be explained by the lower NFC concentration. The lowest pH remained within the minimum limit despite the difference between treatments (*p* < 0.05). The pH remained in the range recommended by the literature for maintaining microbial growth, despite the variation in time collection and among the different roughages.

The pH was influenced by collection time. The minimum estimated pH was 6.38 at 3.79 h after the morning feed supply. When the diet is supplied twice a day, the meals that follow the feed distribution are the most important and last from 1 to 3 h each [34]. Thus, the feed intake per unit of ruminal volume is higher [35], resulting in a drop in ruminal pH soon after the morning diet supply (Table 6).

The highest RAN concentrations observed for NUB and OUB were related to the greater availability of NNP (20 and 19 g/kg DM; Table 3), via urea, with values of 26.68 and 27.73 mg/dL, respectively (Table 6). This concentration of ammoniacal N is essential for microbial growth. The efficiency of N use by microorganisms is directly related to the degradation of fermentable organic matter in the rumen and the available substrate. In tropical conditions, a concentration of N in the rumen environment between 8 and 15 mg/dL is sufficient to intensify the intake and degradation of NDF [36].

There was an interaction for the RAN between the collection time and the different roughages. This result may be related to the quality of the fiber and N sources used in the diets [37]. Microbial synthesis and urea use depend on the digestion rate of carbohydrates, which is the main factor controlling this process’s available energy. [38] stated that an NH_3_ concentration of 23 mg/dL would be optimal to obtain a maximum fermentation rate. Later, [39] determined that the ideal range for maximizing microbial growth is between 10 and 20 mg/dL.

Cellulolytic bacteria use NH_3_ as the primary source of N; however, due to the low fiber quality of bagasse diets, it is impossible to immediately use ammonia in the rumen environment, justifying the higher concentration verified for NUB and OUB.

Typically, the peak ammonia concentration for urea diets is observed about one to two hours after the diet is supplied [35]. The maximum concentrations for NUB, OUB, CS, and SS were 35.4, 41.8, 17.92, and 26.49 (mg/dL), estimated at 2.02, 2.97, 3.01, and 2.87 h after the morning feeding. The NH_3_ concentration of TH (14.33 mg/dL) was constant throughout the collection times. That was the only roughage without urea supplementation.

The interaction between different roughage and collection times on the acetate:propionate ratio was probably due to the higher concentrations of NFC, which are rapidly fermentable in the rumen, present in the NUB and OUB, leading to higher ruminal concentrations in less time. The propionate concentration was practically double for NUB and OUB compared to other roughages. The increase of acetate was slight, which led to a lower acetate:propionate ratio.

Despite showing similarities in the VFA profile between the studied cacti (Table 6), the higher VFA concentrations (μMol/mL) observed for the OUB can be attributed to the more significant number of total sugars present in it (Table 2) compared to NUB. The VFA concentrations increase, and the ruminal pH decreases [33] due to the high fermentation rate of sugars, as verified in the present study. The OUB provided a lower ruminal pH (6.32; Table 6) and higher molar concentration for acetate and propionate (Figure 2). A similar result was verified by Reference [40], showing that the total concentration of VFAs increased with ruminal doses of sucrose or lactose compared to starch-rich diets.

The NUB has a higher starch content (29.0 g/100 g; Table 2) than OUB, despite having a high percentage of fast-digesting carbohydrates (Table 2). According to Reference [40], a slower fermentation rate for starch is expected compared to sucrose or lactose, and this could justify the behavior observed for NUB.

According to the above, we propose cactus cladodes combined with a low-quality fiber source, such as sugarcane bagasse and an NNP source, as a potential solution to forage shortages to improve production systems in semiarid regions and an alternative to the difficulty of forage production. In addition, the diets formulated with cactus and sugarcane bagasse presented a much lower cost considering the current prices in US dollars for the sugarcane bagasse, cactus cladodes, TH, CS, and SS: 0.12, 0.10, 0.27, 0.24, and 0.23 per kg of DM, respectively.

## 5. Conclusions

We recommend using cactus cladodes combined with sugarcane bagasse and urea in sheep diets due to the high degradation rate, greater energy intake, little change in rumen pH, greater volatile fatty acid production, and feasibility, being an alternative to the use of conserved fed such as silages or hays.

## Figures and Tables

**Figure 1 animals-12-03182-f001:**
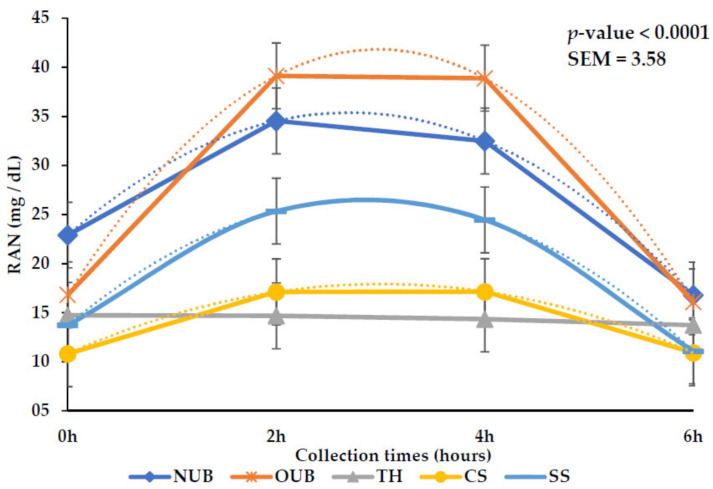
Effect of roughage × collection time interaction (*p* < 0.01) on rumen ammonia nitrogen (RAN) concentration. NUB: y = −1.7088x^2^ + 9.2325x + 22.91; OUB: y = −2.8188x^2^ + 16.793x + 16.82; CS: y = −0.7837x^2^ + 4.7175x + 10.82; SS: y = −1.5606x^2^ + 8.9253x + 13.73.

**Figure 2 animals-12-03182-f002:**
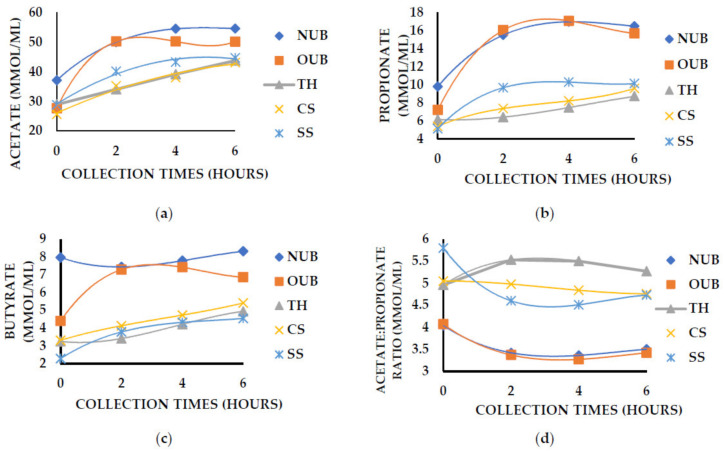
Effect of roughage × collection time interaction (*p* < 0.01) on ruminal concentration (μMol/mL) of acetate (**a**), propionate (**b**), and butyrate (**c**) and acetate:propionate (**d**) ratio.

**Table 1 animals-12-03182-t001:** Chemical composition of the ingredients used in the experimental diets (g/kg DM unless otherwise stated).

Item	*Nopalea*	*Opuntia*	Sugarcane Bagasse	Tifton Hay	Corn Silage	Sorghum Silage	Ground Corn	Soybean Meal
Dry matter, g/kg fresh matter	116	122	911	838	249	230	879	888
Organic matter	876	887	954	914	940	917	983	929
Crude protein	34.0	40.0	11.0	92.0	89.0	60.0	76.0	497
Indigestible crude protein	9.2	9.3	8.5	39.5	12.7	13.1	1.22	138
Neutral detergent fiber	260	302	823	728	590	676	142	141
Indigestible neutral detergent fiber	97.0	119	456	296	181	229	16.0	14.0
Acid detergent fiber	17.2	17.8	64.2	44.8	42.6	46.9	3.9	8.4
Non-fiber carbohydrates	568	532	114	89.0	245	164	724	279

**Table 2 animals-12-03182-t002:** Sugar and starch concentration of cactus cladodes.

Item	*Nopalea*	*Opuntia*
	(g/100 g)	
Fructose	3.84	6.17
Glucose	3.40	4.30
Sucrose	2.09	2.24
Total sugars	8.56	12.7
Starch	29.0	18.8

**Table 3 animals-12-03182-t003:** Proportion of ingredients and chemical composition of the experimental diets.

Item	Diet ^1^				
	NUB	OUB	TH	CS	SS
Ingredients (g/kg DM)					
Cactus *Nopalea*	379	--	--	--	--
Cactus *Opuntia*	--	375	--	--	--
Tifton hay	--	--	694	--	--
Corn silage	--	--	--	692	--
Sorghum silage	--	--	--	--	683
Ground corn	175	175	175	175	175
Soybean meal	115	115	115	115	115
Sugarcane bagasse	295	300	--	--	--
Urea + ammonium sulfate ^2^	20	19	--	02	11
Mineral mix	16	16	16	16	16
Diet composition (g/kg of DM)					
Dry matter ^3^	253	267	853	323	300
Organic matter	893	898	913	929	906
Crude protein	140	139	134	137	141
Ether Extract	15.3	15.7	12.3	18.9	20.2
Neutral detergent fiber	382	401	546	449	503
Indigestible neutral detergent fiber	176	185	209	129	161
Non-fiber carbohydrate	386	375	220	374	283
Total digestible nutrients	712	664	582	641	626

^1^ NUB = *Nopalea* + urea + sugarcane bagasse; OUB = *Opuntia* + urea + sugarcane bagasse; TH = Tifton hay; CS = corn silage; SS = sorghum silage. ^2^ Urea + ammonium sulfate (9:1); ^3^ g/kg as fed.

**Table 4 animals-12-03182-t004:** Mean values for nutrient intake and digestibility in sheep.

Item	Diet ^1^					SEM	*p*-Value
	NUB	OUB	TH	CS	SS		
Intake (g/day)							
Dry matter	1024a	888ab	993ab	983ab	782b	51.1	0.03
Organic matter	904a	792ab	907a	913a	706b	46.3	0.03
Digestible organic matter	670a	550ab	594ab	614ab	469b	39.7	0.04
Crude protein	161a	137ab	145ab	142ab	117b	19.8	<0.01
Neutral detergent fiber	334bc	310c	525a	422ab	381bc	24.3	<0.01
Non-fiber carbohydrate	433a	361ab	209c	327b	206c	18.5	<0.01
Dry matter (g/kg of BW)	26.4ab	24.3ab	27.1a	26.0ab	20.3b	38.80	0.02
Digestibility (g/kg of the nutrient intake)							
Dry matter	712	668	642	672	655	48.4	0.06
Organic matter	741a	692ab	655b	675b	658b	44.0	<0.01
Crude protein	831a	806a	753b	715c	768b	20.3	<0.01
Neutral detergent fiber	561ab	496b	645a	566ab	595ab	24.7	0.01

^1^ NUB = *Nopalea* + urea + sugarcane bagasse; OUB = *Opuntia* + urea + sugarcane bagasse; TH = Tifton hay; CS = corn silage; SS = sorghum silage. BW = body weight. Means followed by different letters on the same line differ by the Tukey test (*p* < 0.05).

**Table 5 animals-12-03182-t005:** Pool sizes and rates of ingestion (Ki), passage (Kp), and digestion (Kd) of DM e NDF, and passage rate of iNDF (Kpi).

Item	Diet ^1^					SEM	*p*-Value
	NUB	OUB	TH	CS	SS		
Pool ruminal (g)							
Dry matter	505ab	408b	593a	448b	421b	25.2	<0.01
Neutral detergent fiber	304ab	251b	384a	292ab	278b	20.6	<0.01
Indigestible neutral detergent fiber	142ab	120b	178a	96.4b	101b	10.42	<0.01
Dry matter (h^−1^)							
Ki	0.0850	0.0930	0.0716	0.0934	0.0810	0.005	0.05
Kp	0.0306	0.0342	0.0356	0.0372	0.0394	0.002	0.20
Kd	0.0548ab	0.0584a	0.0360c	0.0566a	0.0418c	0.001	<0.01
Neutral detergent fiber (h^−1^)							
Ki	0.0462	0.05340	0.0590	0.0616	0.0594	0.003	0.07
Kp	0.0210	0.0236	0.0250	0.0260	0.0268	0.008	0.29
Kd	0.0254b	0.0296ab	0.0344ab	0.0360a	0.0328ab	0.002	0.02
Kpi	0.0426	0.0492	0.0477	0.0541	0.0538	0.004	0.34

^1^ NUB = *Nopalea* + urea + sugarcane bagasse; OUB = *Opuntia* + urea + sugarcane bagasse; TH = Tifton hay; CS = corn silage; SS = sorghum silage. Means followed by different letters on the same line differ by the Tukey test (*p* < 0.05).

**Table 6 animals-12-03182-t006:** Effects of treatment and collection time on concentration of rumen ammonia nitrogen (RAN, mg/dL), volatile fatty acids, and ruminal pH in sheep.

Item	Diet ^1^						*p*-Value		
	NUB	OUB	TH	CS	SS	SEM	Treat	Time	Treat × Time
RAN	26.6a	27.7a	14.3b	14.0b	18.66b	2.81	<0.01	<0.01	<0.01
Volatile fatty acids, VFA (μMol/mL)									
Acetate (A)	49.1	45.3	36.4	35.5	39.2	4.17	<0.01	<0.01	<0.01
Propionate (P)	14.7	14.0	7.22	7.59	8.82	1.04	<0.01	<0.01	<0.01
Butyrate	7.92	6.54	3.87	4.39	3.72	0.83	<0.01	<0.01	<0.01
A:P ratio	3.3	3.2	5.1	4.6	4.4	0.25	<0.01	<0.01	<0.01
pH	6.50b	6.32c	6.65a	6.62ab	6.56ab	0.10	0.02	<0.01	0.73
**Ruminal pH on the collection times (hours)**									
	**0 h**	**2 h**	**4 h**	**6 h**	**EPM**	***p*-Value**			
	6.80	6.44	6.42	6.51	0.088	Linear		Quadratic	
						<0.01		0.03	

^1^ NUB = *Nopalea* + urea + sugarcane bagasse; OUB = *Opuntia* + urea + sugarcane bagasse; TH = Tifton hay; CS = corn silage; SS = sorghum silage. RAN = ruminal ammonia nitrogen. Means followed by different letters on the same line differ by the Tukey test (*p* < 0.05).

## Data Availability

http://www.ppgz.ufrpe.br/sites/default/files/testes-dissertacoes/tese_michelle_siqueira_051120.pdf (accessed on 10 May 2022).

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
