# Peer review of "Cactus Cladodes Opuntia or Nopalea and By-Product of Low Nutritional Value as Solutions to Forage Shortages in Semiarid Areas"

_animals, 2022, doi:10.3390/ani12223182_

Round 1
Reviewer 1 Report
The authors mentioned the significance and feasibility of studying of new components of animal diets (Cactus cladodes Opuntia and Nopalea) in semi-arid areas in order to overcome the problem of lack of feed. In my opinion, this is a relevant and valuable addition for agriculture sector (large and small livestock). Since, the shortage of such feeds as silage and hay, it is possible to replace alternatives more efficient and cheaper feeds, such as cactus. Generally, the work is relevant and of great interest for further development. However, during the reviewing process, some imprecisions and recommendations to the authors were noted, to which we would like to receive a complete answer:
1. In the process of reviewing the manuscript, only one time is mentioned (Line 61-67) on the use of cacti in feeding ruminants. The authors need to describe in more detail about similar studies of cacti in animal feeding conducted by other researchers (from South Africa, Mexico, Morocco, and Tunisia). They should provide information on the nutritional value and chemical composition of cacti (proteins, lipids, minerals, etc.).
2. Had the feed from Cactus cladodes Opuntia and Nopalea been crushed? If yes, to what particle size? The speed of passage of feed through the gastrointestinal tract of animals and, consequently, the digestibility of nutrients depends on the particle size.
3. Was the particle size of other feeds (Sugarcane Bagasse, Tifton Hay, Corn Silage, Sorghum Silage) taken into account when distributing the total mixed ration (TMR)?
4. It is not completely clear from the text of the manuscript in what form cactus feeds (Cactus cladodes Opuntia and Nopalea) were fed. In fresh form (without grinding (whole), with grinding) or in siloed form?
5. How did you make sure to consume Cactus cladodes Opuntia and Nopalea with your diet? Was there any separation of feed when consumed by experimental animals?
6. It was also necessary to explore the content of ADF, RUP (rumen undegradable protein) and DP (degradable protein) in feed and diet (table 1 and 3).
7. On what basis or recommendations or programs the diets were calculated? Also, according to the content of mineral substances, the diet was balanced?
8. In Table 3 (Line 117), the decoding of "Urea/ as" to "Urea + ammonium sulfate" follows.
9. Have you conducted a study of the health status of experimental animals when starting feeding Cactus cladodes Opuntia and Nopalea?
10. Line 136 and 137: for what purpose do you give the equation for calculating the coefficients of digestibility, if you do not apply it further in the text of the manuscript? The digestibility of nutrients is usually measured either in digestibility coefficients (%) or in the amount of digestible nutrients (g) (in the table g / kg of natural feed, consumed feed or dry matter?). Should you give a reference or link to the literature source from where this equation was taken (I think it is not correct)?
11. It is better to insert figures and drawing in the manuscript in color form, because in the current form (black and white) it is difficult to navigate (Line 233, 250, 251).
12. Was the content of mineral elements in the diets the same between the groups? The composition of the premix in all groups was the same?
13. Line 139 and 141: only the diet, feed or leftovers were crushed, it is not entirely clear what the authors wanted to say (please be more concise)?
14. Line 117: Instead of "00" should be put "–".
15. On the basis of what technique was the ruminal fluid preserved (give a reference)? Have you analyzed the total amount of volatile fatty acids (VFA) in the ruminal content or not?
16. The discussion should be divided into sub-chapters (similar to the research methodology "Intake and digestibility of nutrients", "Neutral detergent fiber and dry matter and ruminal dynamic", "Parameters of Ruminal fermentation"). This correction will be better for readers' awareness.
17. The subtitle "3.3 Ruminal parameters" is better to make "3.3 Indicators of ruminal fermentation".
Author Response
We would like to thank you for the relevant comments and points addressed for this review. Please, find the answers below and highlighted in the text in accordance with the Line (L) description.

Reviewer 2 Report
Some suggestions are offered to the authors.
I kindly suggest detailing all the handling of the cladodes from harvest to incorporation into diets.
The cladodes had spines and had to be cleaned or were they the forage type?
After the cladodes were harvested, were they chopped, sun-dried or what was their preparation to incorporate them into the experimental diets?
How does the clsdodios were offered? Fresh chopped?
What was the average particle size of the cladodes to incorporate them into the diets?
How the diets were mixed. By hand using shovels or mechanical mixers?
Was there selectivity in the different experimental diets? This is important for calculating nutrient intake and analysis of digestibility results.
Cladodes used for animal feed can be consumed by humans?
It would be interesting to carry out a cost-benefit analysis where the entire process is included, from harvest to incorporation into the diets, of the cladodes.
Author Response

(The authors gave the same response as above.)

Round 2
Reviewer 1 Report
The article has been substantially revised. Recommended for further publication.